# Do Transferrin Levels Predict Haemodialysis Adequacy in Patients with End-Stage Renal Disease?

**DOI:** 10.3390/nu11051123

**Published:** 2019-05-20

**Authors:** Giovanni Tarantino, Mauro Vinciguerra, Annalisa Ragosta, Vincenzo Citro, Paolo Conforti, Giovanni Salvati, Aniello Sorrentino, Luca Barretta, Clara Balsano, Domenico Capone

**Affiliations:** 1Department of Clinical Medicine and Surgery, “Federico II” University Medical School of Naples, 80131 Naples, Italy; 2Section of Nephrology, “Santa Maria Delle Grazie” Hospital, 80078 Pozzuoli, Italy; mauro.vinciguerra@virgilio.it; 3Outpatients Clinic of Hemohaemodialysis Dial Center s.r.l., Pomigliano D’Arco, 80038 Naples, Italy; Annalisaragosta@libero.it (A.R.); Giosal1955@libero.it (G.S.); 4Department of General Medicine, “Umberto I” Hospital, Nocera Inferiore, 84014 Salerno, Italy; v.citro@libero.it; 5Medical School of Naples, “Federico II” University, 80131 Naples, Italy; pconforti8@gmail.com; 6Diagnostic Center Kappa s.r.l., Pomigliano D’Arco, 80038 Naples, Italy; info@kappa-srl.it (A.S.); info@centrodiagnosticokappa.it (L.B.); 7Department of Clinical Medicine, Life, Health & Environmental Sciences-MESVA, University of L’Aquila, 67100 L’Aquila, Italy; clara.balsano@cc.univaq.it; 8Integrated Care Department of Public health and Drug Use, Section of Medical Pharmacology and Toxicology, “Federico II” University, 80131 Naples, Italy; docapone@unina.it

**Keywords:** nutritional status, inflammation, iron status, haemodialysis adequacy

## Abstract

Background: Since haemodialysis is a lifesaving therapy, adequate control measures are necessary to evaluate its adequacy and to constantly adjust the dose to reduce hospitalisation and prolong patient survival. Malnutrition is common in haemodialysis patients and closely related to morbidity and mortality. Patients undergoing haemodialysis have a high prevalence of protein-energy malnutrition and inflammation, along with abnormal iron status. The haemodialysis dose delivered is an important predictor of patient outcome. Aim: To evaluate through haemodialysis adequacy, which parameter(s), if any, better predict Kt/V, among those used to assess nutritional status, inflammation response, and iron status. Methods: We retrospectively studied 78 patients undergoing haemodialysis due to end-stage renal disease. As parameters of nutritional status, geriatric nutritional risk index (GNRI), transferrin levels, lymphocyte count, and albumin concentration were analysed. As signs of inflammation, C reactive protein (CRP) levels and ferritin concentrations were studied as well. Iron status was evaluated by both transferrin and ferritin levels, as well as by haemoglobin (Hb) concentration. Results: The core finding of our retrospective study is that transferrin levels predict the adequacy of haemodialysis expressed as Kt/V; the latter is the only predictor (*P* = 0.001) when adjusting for CRP concentrations, a solid marker of inflammation, and for ferritin levels considered an iron-storage protein, but also a parameter of inflammatory response. Discussion and Conclusion: In keeping with the results of this study, we underline that the use of transferrin levels to assess haemodialysis quality combine into a single test the evaluation of the three most important factors of protein-energy wasting.

## 1. Introduction

Ever since haemodialysis therapy has proven to be a lifesaving therapy, it has become essential to evaluate the adequacy of treatment and to constantly adjust the dose to meet the goals of the Kidney Disease Outcomes Quality Initiative [1]. To this regard, clinicians have been challenged to reduce hospitalisation and prolong the survival of haemodialysis patients through adequate control measures.

Malnutrition is common in haemodialysis patients and closely related to morbidity and mortality. Therefore, the assessment of nutritional status and, consequently, the nutritional management of these patients play a central role [2].

However, since malnutrition also occurs in pre-haemodialysis patients, it is evident that other factors unrelated to this condition (e.g., infectious and inflammatory complications, as well as co-morbidities), may also be important contributors to malnutrition in end-stage renal disease (ESRD) [3]. Recently, experts have labelled uremic malnutrition as a protein-energy wasting (PEW) condition [4]. The diagnosis of PEW is based on four main diagnostic criteria: (a) laboratory tests—such as serum albumin, prealbumin, transferrin, and cholesterol; (b) measures of body mass—body mass index (BMI), unintentional weight loss, and total body fat; (c) measures of muscle mass—total muscle mass, mid-arm muscle circumference, and creatinine clearance; (d) measures of dietary protein and energy intake. Additional parameters that should be taken into account are specific markers of inflammation as well as integrative nutritional scoring systems [5]. The limitations of each of these parameters are discussed below.

Although commonly used to evaluate haemodialysed patients, albumin levels as a marker of nutritional status are somewhat criticised because they are affected by inflammation and hydration status [6]. An interesting parameter is lymphocyte count, also expressed as a percentage of total white blood count (WBC), which may be related to the clinical outcome in maintenance haemodialysis (MHD) patients, inasmuch as it is a valid predictor of mortality [7]. Actually, some doubt has been cast on its reliability as a nutritional marker alone, since it can also indicate an ongoing inflammatory process. As previously stressed, patients undergoing MHD have a high prevalence of protein-energy malnutrition and inflammation. Since these two conditions often occur concomitantly in MHD patients, they are referred to the ‘malnutrition–inflammation complex syndrome’ (MICS) [8].

As to malnutrition, mid-arm muscle circumference provides an estimate of somatic protein reserve. This non-invasive and inexpensive tool is an early indicator of nutritional depletion. However, the associated loss of muscle strength and impaired mobility (sarcopenia can be observed in chronic debilitating diseases, like rheumatoid arthritis or in the proximity of injured joints, as in the case of osteoarthritis), weight loss and lung function have untoward consequences on this measurement [9].

In relation to other anthropometric measurements, BMI cannot distinguish between lean mass and fat or bone. To overcome this gap, a recently developed measurement of body composition provides a possibly objective assessment, but requires expensive equipment and adequate expertise. The techniques include dual-energy X-ray absorptiometry (DXA, previously DEXA), stable isotope of thetechniques (D2O dilution), and bioelectrical impedance analysis (BIA). BIA provides a reliable estimate of total body water in most conditions and therefore can be a useful tool for body composition analysis. Nonetheless, BIA values are affected by numerous variables including body position, hydration status, consumption of food and beverages, ambient air and skin temperature, recent physical activity, and conductance of the examining table. Reliable BIA requires standardisation and control of these variables. Measurements obtained include body fat mass (FM), and lean body mass (LBM) [10].

The geriatric nutritional risk index (GNRI), calculated on the basis of serum albumin and body weight, is a simple method for predicting nutritional status and clinical outcome in haemodialysis patients [11]. Based on recent evidence, in addition to elderly age, malnutrition (low GNRI), and inflammation (high ferritin concentrations) have been identified as significant independent risk factors that predict all-cause mortality in haemodialysis patients [12].

Additionally, patients with ESRD can present as having functional iron deficiency; this is characterised by the presence of sufficient iron stores (as per conventional criteria, i.e., ferritin serum levels) that cannot however be mobilized to ameliorate erythropoiesis through the administration of erythropoietin. These patients have either normal or high serum ferritin levels, but present a frankly decreased transferrin saturation.

It is therefore of paramount importance to screen functional iron deficiency—which usually responds to iron therapy—from the so-called inflammatory iron block, which is insensible to supplementation, due to an underlying inflammatory state characterised by increased C reactive protein (CRP), an acute-phase reactant linked to augmented cardiovascular disease risk [13]. In addition, transferrin serum concentrations or total iron-binding capacity, linearly correlated with transferrin, is also a good marker of nutritional status, although low levels in poorly nourished patients may have implications when diagnosing and treating anaemia and iron deficiency in malnourished haemodialysis patients [14]. It is important to stress that transferrin is also known as negative acute phase protein [15].

The dose of haemodialysis delivered is an important predictor of patient outcome (i.e., mortality) [16]. Two methods are generally used to assess haemodialysis adequacy, urea reduction ratio (URR), and Kt/V. The latter is more accurate than URR in measuring the amount of urea removed during haemodialysis, primarily because Kt/V also considers the urea eliminated with excess fluid [17]. Nevertheless, this urea-based standard (Kt/V) is not always a useful tool to assess better patient outcome or more favourable measures of health-related quality of life—an issue of concern namely for patients—since other factors such as inflammation parameters, nutritional status evaluation, hemodynamic assessment, and volume control are of clinical cardinal importance [18].

Indeed, there is ongoing controversy about the correct use of the most suitable parameters to assess nutritional status and inflammation response as well as iron status. Furthermore, there is a need to weigh dialysis adequacy because it is independently correlated with mortality. In keeping with this, we aimed at evaluating which factor(s), if any, better predicted the Kt/V.

## 2. Methods

### 2.1. Study Design

This is a retrospective study based on data collected from patients’ records; the findings influenced in no way the decision-making of physicians, who strictly followed the diagnosis and therapy guidelines in caring for their haemodialysed patients.

### 2.2. Population

We analysed data from 105 ESRD patients who had not experienced chronic stable angina for at least three months and who were without a history or interventional assessment of ischemic heart disease in the same period. Changes in electrocardiogram responses to exercise (treadmill heart test) were assessed in all subjects to rule out cardiac ischemic signs.

In relation to the clinical characteristics of our population of dialysed patients (pts), it is noteworthy to stress that ESRD was due to type 2 diabetes mellitus (35 pts), polycystic disease (4 pts), glomerulonephritis (6 pts), hypertension (7 pts), cancer (2 pts), Alport syndrome (1 pt), unknown origin (23 pts), and as a comorbidity (6 pts) of chronic liver diseases.

### 2.3. Exclusion Criteria

On the basis of the suspected or verified diagnoses reported in the medical records examined, we excluded 15 patients with chronic diseases that could have induced malnutrition/weight loss due to digestive disorders: lower vitamin B-12 levels (atrophic gastritis) (1 pt); IgA anti-tissue transglutaminase antibodies (celiac disease) (2 pts); intestinal symptoms such as diarrhoea or fever of unknown origin and recurrent severe abdominal pain (Crohn’s disease) (1 pt); ultrasound features (chronic pancreatitis) (3 pts); altered BMI due to disrupted thyroid gland function (3 pts). Alcohol abuse was assessed following the DSM-IV diagnostic criteria, by means of screening tests such as MAST (Michigan Alcohol Screening Test) and CAGE (Cut down, Annoyed, Guilty, and Eye opener), as well as random tests for blood alcohol concentration and the use of a surrogate marker (e.g., mean corpuscular volume (5 pts), potential insufficient calorie intake (3 pts on lipid-lowering diet for dyslipidaemia, 4 pts on hypoglycemic diet for impaired glucose tolerance, 2 pts for dyspepsia), unsatisfactory recording of clinical data (3 pts). Finally, in this retrospective study, 78 ESRD patients were considered. 

### 2.4. Laboratory Data

CRP measurements were performed by Tina-quant C Reactive Protein Gen.3 (Roche Diagnostics GmbH, Nonnenwald 2 D-82377, Penzberg, Germany). Using Cobas instruments, ferritin was determined by an electro-chemiluminescent immunoassay, transferrin and albumin were evaluated by an immunoturbidimetric assay, and urea levels by a kinetic method. Haemoglobin (Hb) concentration, white blood count (WBC), and lymphocyte count were obtained by impedance measurement with an Advia 2120 instrument. GNRI was calculated on the basis of the following formula [19]: GNRI = (1.489 × albumin (g/L)) + (41.7 × (body weight/ideal body weight)). Ideal body weight was calculated using the Lorentz equation [18]. Kt/V was estimated by using the suggested equation [20]: Kt/V = −ln(Ct/C0 − 0.008T) + (4 − 3.5 × Ct/C0) × BW/BWt, where Ct and C0 = post- and pre-haemodialysis plasma urea nitrogen levels (mg/dL); T = haemodialysis session length (hours); BW = haemodialysis ultrafiltration (kg); BWt = post-haemodialysis body weight (kg). Initial serological testing for chronic hepatitis B virus (HBV) and hepatitis C virus (HCV) infection was conducted using either rapid diagnostic tests (RDT) or laboratory-based enzyme immunoassays (EIAs) for the detection of hepatitis B surface antigen (HBsAg) or antibodies to HCV (anti-HCV).

### 2.5. Statistics

Data were derived from a normally distributed population and are given as mean plus standard deviation (SD). Variables not normally distributed were expressed as median plus 25–75 interquartile range (IQR). Frequencies were studied by the chi-squared test.

To study predictions we chose various types of regression techniques. To this regard, the predicted values do not depend on the order of predictors in the equation, as we are always solving the same equation. At univariate analysis, the linear regression analysis (ordinary least squares (OLS)) was used evaluating the coefficient with its standard error, 95% confidence intervals (CI), the t (t-value), and R-squared. Understanding R-squared boils down to grasping the separate concepts of central tendency and variability, and how they relate to the distribution of data points around the fitted line. When heteroscedasticity was suspected (i.e., in case of sub-populations with different variabilities from others in the homoscedastic model), having detected the presence of few outliers, we analysed the correlation by the robust regression, using least absolute deviations (LAD) regression.

At multivariate analysis, the multivariate regression was added by using as a dependent variable the adequacy of haemodialysis and various parameters as predictors, in order to study which of the independent variables had the greatest effect on the dependent variable.

## 3. Results

The main characteristics of our population are summarised in Table 1.

Interestingly, no gender difference emerged, as there were 36 females versus 42 males (chi-squared, *P* = 0.33). The median Hb concentration in our patients was 11.1 (10.45–11.6) g/dL in females and 11.3 (10.4–12.1) g/dL in males, confirming the presence of anaemia. The median GNRI value of our population was 100.7 (96.1–106.9), being only the median of the first quartile below 98, the cut-off value for no risk [21]. The Kt/V value was adequate, being 1.41 ± 0.23. According to US guidelines, for three times a week haemodialysis, a Kt/V (without rebound) should be at least 1.2 with a target value of 1.4 (15% above the minimum values) [22].

### Predictions

The first step consisted in evaluating predictions of the GNRI by other parameters used to evaluate the nutritional status (i.e., lymphocyte count and transferrin levels). Furthermore, we studied the prediction of GNRI by iron status and levels of the main marker of inflammation (i.e., CRP). Finally, we sought to understand whether the adequacy of haemodialysis could be predicted by the GNRI. Noteworthy, the GNRI did not predict Kt/V in our population.

The outputs are shown in Table 2.

Being CRP concentration canonically linked to cardiovascular diseases and, consequently, to all-cause mortality, we also assessed the relationship between this marker of inflammation and both nutritional status and iron status on the one hand, and the quality of haemodialysis on the other. Intriguingly, no evident prediction was found in terms of quality of haemodialysis, whereas CRP concentrations negatively predicted only three parameters of nutritional status, (i.e., GNRI, albumin and lymphocyte count) but not transferrin levels, as shown in Table 3.

To test the importance of lymphocyte count, we further studied the possible relationship between this parameter, considered as marker of nutritional status, but also of inflammation (this was confirmed considering its prediction by CRP) with two other parameters commonly used to assess nutritional status (i.e., transferrin and albumin levels). The output summarised in Table 4 suggests that malnutrition and inflammation are intertwined but follow divergent paths.

The lack of prediction of both transferrin and ferritin levels by CRP concentrations prompted us to evaluate their association, which led to an expected result, as evident in Table 4.

Furthermore, we evaluated the link between the aforementioned parameters of iron status and haemoglobin concentration, to try to understand the role of the suspected iron blockade.

Interestingly, a link was detected only between ferritin and transferrin levels, as shown in Table 5.

As previously reported, a noticeable finding is the lack of relationship between the quality of haemodialysis evaluated as Kt/V and by the GNRI. Nevertheless, to further assess this aspect (i.e., the role of nutritional status in evaluating the adequacy of haemodialysis), we explored a possible association between other parameters of MICS presence, but also of iron status and Kt/V, the results of which are presented in Table 6.

Interestingly, cholesterol levels, a marker used as diagnostic criterion in the PEW, did not predict dialysis adequacy (Coef., Standard Error, t, P > |t| and 95%CI: 0.00071, 0.0006836, 1.04, 0.302, and −0.0006519/0.0020718, respectively).

As a core finding, we ascertained the superiority of transferrin in predicting haemodialysis adequacy. At multivariate regression using Kt/V as a dependent variable, and CRP, transferrin, and ferritin (variables that at univariate analysis showed prediction), the only one that remained able to predict Kt/V was transferrin serum concentration, as clearly shown in Table 7.

Since Kv/T is not widely accepted as an outcome, we generated a new model using the GNRI as a marker of nutritional status, comparing it with transferrin, CRP and Kt/V (Table 8).

## 4. Discussion

To better understand the complex interplay between inflammation and malnutrition (with iron status playing a relevant role), which characterises the so-called PEW, it is relevant to point out the inherent vicious cycle due to a sequence of reciprocal cause and effects where three elements (i.e., inflammation, malnutrition, and iron blockade) intensify and aggravate each other, inexorably worsening the situation.

The core finding of our retrospective study is that transferrin levels predict the adequacy of haemodialysis expressed as Kt/V. It remains the only predictor when adjusted for CRP concentration, a solid marker of inflammation and ferritin levels of the iron-storage protein, but is also recognised as a parameter of inflammatory response.

Measurement of transferrin has been always considered a rapid way to estimate iron supply. However, this parameter has wide fluctuations due to the serum iron levels and is dependent on the individual’s diet (malnutrition). On the basis of our results, another important advantage of transferrin measurement is its application in haemodialysis patients, possibly showing inflammation, which is often unidentifiable by clinical examination, but quite common in haemodialysis. In keeping with the proposed diagnostic advantage of this parameter are findings from the clinical investigation.

The importance of transferrin is clearly evidenced by a milestone study, in which both multivariate Poisson and Cox models adjusted for demographic features, haemodialysis dose and age, serum albumin, ferritin, blood haemoglobin concentrations, administered epoietin and iron doses, showed that serum iron levels as well as iron saturation ratios were significantly, but inversely, associated with prospective mortality and hospitalisation in haemodialysis patients. Interestingly, this inverse association remained significant in a sub-cohort of 322 patients on maintenance haemodialysis after additional adjustments for co-morbidities and serum CRP levels, reflecting inflammation [23].

As collateral findings, we stress that ferritin and transferrin levels, markers of both inflammation and iron status, did not impact on the ESRD-linked anaemia in this population. The ferritin concentrations, both in males and females, were not particularly high, falling in most cases within the normal range, thus indicating sufficient iron stores.

As a collateral finding, we should point out that ferritin levels among females were substantially above the normal range, while in males they were moderately elevated, showing a clear gender difference and highlighting the role of ferritin as a marker of inflammation more than as an indicator of iron status [24].

At the same time, lymphocyte count did not predict serum albumin or transferrin levels, likely playing a secondary role as nutritional status parameter.

In relation to the similarities and discrepancies of our results compared to other research, we should underline the broad ongoing debate on various matters.

First, the measurement of haemodialysis quality remains a difficult issue. Currently, over 93% of US patients reach the recommended target Kt/V urea; nonetheless, the survival rate of haemodialysis patients remains unsatisfactory: the mortality of patients on MHD is fourfold higher than in the general population aged >65 years [25].

This observation has prompted experts to cast some doubts on the validity of Kt/V as a perfect tool, although this urea-based standard provides a useful tool to avoid grossly inadequate haemodialysis [18,26].

Secondly, the causes of inflammation are multifactorial and include patient-related factors such as underlying diseases, comorbidity, nutritional status, oxidative stress, infections, obesity, and genetic or immunologic factors, but also a factor linked to haemodialysis technique, which may contribute to maintaining the inflammatory state in many haemodialysed patients [27].

Thus, evaluating only patient-related components may not provide the complete scenario. As a possible strength of this study, we emphasise that the detection of transferrin levels to assess haemodialysis quality combines in a single test the evaluation of three of the most important factors of PEW. Furthermore, as both low-dose haemodialysis (Kt/V) and small body size were found to be independent risk factors for mortality in chronic haemodialysed patients, a parameter evaluating haemodialysis adequacy could improve the management of these patients and, consequently, their health [28].

An evident weakness, the authors acknowledge the lack of information on these patients’ intima-media thickness parameters to detect evidence of early atherosclerosis and, therefore, of the so-called malnutrition-inflammation-atherosclerosis (MIA) syndrome [29].

Noteworthy, we did not measure serum hepcidin, whose reduced circulating concentrations due to haemodialysis and/or epoietin treatment may positively affect erythropoiesis [30]. Nor did we evaluate iron and ferritin ratio, which not only reflects the bioavailability of iron for the body, but also a healthy body composition [31].

Several markers have been used to detect inflammatory reactions in patients with chronic renal disease. These can be used to predict future cardiovascular events. Among the several parameters of inflammatory markers (e.g., TNF-alpha and Interleukin-6), serum CRP is well known and its advantages for the detection of inflammation and its predictive potential have been evaluated in several studies. In addition, its ability in predicting future atherosclerosis and the effects of treatment are noteworthy. In our study, we dosed high sensitivity C-reactive protein (hsCRP), a more precise marker that seems to contribute to the development of cardiovascular complications [32]. As these findings are preliminary, they should be taken with due caution, pending more conclusive evidence.

### Limitations

This study, on the whole, has a variety of intrinsic drawbacks, including the fact that many patients lacked a comprehensive record of their caloric intake (unclear risk of bias). Secondly, the lack of follow-up periods was due to the substantially reduced life expectancy of these patients [33]. Furthermore, the small size of our cohort does not allow us to draw strong conclusions, although these results should be considered as a starting point for larger studies. Again, it did not evaluate more in depth the role of iron deficiency, studying the favourable effects of intravenous iron on the functional status and quality of life in patients undergoing MHD, in light of its cardio-protective action [34]. Finally, haemodialysis adequacy, evaluated as Kt/V, remains an open issue and whether it could be used as an indirect outcome.

## 5. Conclusions

The preliminary results of this observational study, albeit limited by its retrospective design, show that assessing haemodialysis quality by detecting transferrin levels, combines in a single test the evaluation of three of the most important factors of protein-energy wasting. Nevertheless, given the complexity of factors affecting both nutritional status and iron status, larger prospective studies are warranted to confirm our data.

## Figures and Tables

**Table 1 nutrients-11-01123-t001:** Demographic and laboratory characteristics of 78 end-stage renal disease (ESRD) patients.

Age (Years)	68 (64–77)	Gender (M/F) *n*	42/36
Dialysis age (months)	41.5 (13–70)	CRP mg/L	10.4 (2–23.5)
Ferritin ng/dL (M) Ferritin ng/mL (F)	179 (91–319) 253/(190–376)	Transferrin mg/dL	187.5 ± 39.1
Iron supplementation (yes/no) *n* (%)	33 (46.48) 38 (53.52)	Albumin g/L	3–6 (3.34–3.81)
Lymphocytes %	18.5 ± 5–9	Hb (M) g/dL Hb (F) g/dL	11.3 (10.4–12.1 11.1 (10.45–11.6)
Cholesterol	130.5 (115–153)	GNRI	100.7 (96.1–106.9)
Kt/V	1.41 ± 0.23	BMI	26.5 (24.6–28.8)

Legend to Table 1. M, males; F, females; ESRD, end-stage renal disease, CRP, C reactive protein, Hb, haemoglobin, GNRI, geriatric nutritional risk index, BMI, body mass index.

**Table 2 nutrients-11-01123-t002:** Predictions between geriatric nutritional risk index (GNRI) and lymphocyte count, transferrin levels, the iron status, the main marker of inflammation, and the adequacy of haemodialysis.

Linear regression, Robust	Number of obs = 78		R-squared = 0.0759		
d.v. GNRI	Coefficient	Standard Error	T	P > |t|	95% Conf. Interval
i.v. Transferrin	0.0824125	0.028789	2.86	0.005	0.0250742/.1397508
**Linear regression, Robust**	**Number of obs = 74**		**R-squared = 0.0642**		
d.v. GNRI	Coefficient	Standard Error	T	P > |t|	95% Conf. Interval
i.v. Transferrin	0.4786761	0.2180616	2.20	0.031	0.0439782/.913374
**Linear regression, Robust**	**Number of obs = 69**		**R-squared = 0.0152**		
d.v. GNRI	Coefficient	Standard Error	T	P > |t|	95% Conf. Interval
i.v. Lymphocyte	−0.0080178	0.0065333	−1.23	0.224	−0.0210583/0.0050228
**Linear regression, Robust**	**Number of obs = 77**		**R-squared = 0.0825**		
d.v. GNRI	Coefficient	Standard Error	T	P > |t|	95% Conf. Interval
i.v. CRP	−0.1155926	0.0347156	−3.330	0.001	−0.1847496/−0.0464355
**Linear regression, Robust**	**Number of obs = 78**		**R-squared = 0.0013**		
d.v. Kt/V	Coefficient	Standard Error	T	P > |t|	95% Conf. Interval
i.v. GNRI	0.0007452	0.0024208	0.31	0.759	−0.0040762/0.0055667

Legend to Table 2. Significant predictions are represented in bold; d.v., dependent variable; i.v., independent variable. The low R-squared in the presence of significance shows that even noisy, high-variability data can have a significant trend.

**Table 3 nutrients-11-01123-t003:** Prediction of parameters studying nutritional status, iron status, and quality of haemodialysis by C reactive protein (CRP) levels.

Linear regression, Robust	Number of obs = 77		R-squared = 0.0825		
d.v. GNRI	Coefficient	Standard Error	T	P > |t|	95% Conf. Interval
i.v. CRP	−0.1155926	0.0347156	−3.33	0.001	−0.1847496/−0.0464355
**Linear regression, Robust**	**Number of obs = 77**		**R-squared = 0.0948**		
d.v. Albumin	Coefficient	Standard Error	T	P > |t|	95% Conf. Interval
i.v. CRP	0.0037217	0.0010437	−3.57	0.001	−0.0058008/−0.0016426
**Linear regression, Robust**	**Number of obs = 77**		**R-squared = 0.0161**		
d.v. Transferrin	Coefficient	Standard Error	T	P > |t|	95% Conf. Interval
i.v. CRP	−0.1725484	0.2506418	−0.69	0.493	−0.6718526/−0.3267558
**Linear regression, Robust**	**Number of obs = 73**		**R-squared = 0.0620**		
d.v. Lymphocyte	Coefficient	Standard Error	T	P > |t|	95% Conf. Interval
i.v. CRP	−0.0520072	0.0137425	−3.78	0.000	−0.079409/−0.0246054
**Linear regression, Robust**	**Number of obs = 68**		**R-squared = 0.0007**		
d.v. Ferritin	Coefficient	Standard Error	T	P > |t|	95% Conf. Interval
i.v. CRP	0.1561528	0.8402418	0.19	0.853	1.521444/1.83375
**Linear regression, Robust**	**Number of obs = 77**		**R-squared = 0.0232**		
d.v. Kt/V	Coefficient	Standard Error	T	P > |t|	95% Conf. Interval
i.v. CRP	−0.0012742	0.001169	−1.09	0.279	−0.0036029/−0.0010544

Legend to Table 3. Significant predictions are represented in bold; d.v., dependent variable; i.v., independent variable. The low R-squared in the presence of significance shows that even noisy, high-variability data can have a significant trend. Of note is the prediction of lymphocyte count by CRP levels, confirming its validity in exploring the inflammatory status.

**Table 4 nutrients-11-01123-t004:** Prediction of albumin and transferrin levels by lymphocyte count.

Linear regression, Robust	Number of obs = 74		R-squared = 0.0093		
d.v. Albumin	Coefficient	Standard Error	T	P > |t|	95% Conf. Interval
i.v. Lymphocyte	0.0054099	0.006815	0.79	0.430	−0.0081756/−0.096219
**Linear regression, Robust**	**Number of obs = 74**		**R-squared = 0.018**		
d.v. Transferrin	Coefficient	Standard Error	T	P > |t|	95% Conf. Interval
i.v. Lymphocyte	0.268882	0.74110190.36	0.36	0.718	−1.208478/1.746242.

Legend to Table 4. Noteworthy is the lack of prediction of the well-accepted parameters of nutritional status by lymphocyte count, casting some doubts on its validity in exploring the nutritional status alone, but more likely useful in evaluating the so-called malnutrition–inflammation complex syndrome (MICS); d.v., dependent variable; i.v., independent variable.

**Table 5 nutrients-11-01123-t005:** Prediction of parameters directly or indirectly evaluating iron status.

Linear regression, Robust	Number of obs = 69		R-squared = 0.1726		
d.v. Transferrin	Coefficient	Standard Error	T	P > |t|	95% Conf. Interval
i.v. Ferritin	−0.0965767	0.0252629	−3.82	0.000	−0.1470016/−0.0461518
**Linear regression, Robust**	**Number of obs = 69**		**R-squared = 0.0019**		
d.v. Hb	Coefficient	Standard Error	T	P > |t|	95% Conf. Interval
i.v. Ferritin	−0.0003092	0.0008286	−0.37	0.710	−0.001963/−0.0013446
**Linear regression, Robust**	**Number of obs = 78**		**R-squared = 0.0119**		
d.v. Hb	Coefficient	Standard Error	T	P > |t|	95% Conf. Interval
i.v. Transferrin	0.0032726	0.0031433	1.04	0.301	−0.0029878/0.0095331

Legend to Table 5. Hb, haemoglobin concentration. The negative prediction of transferrin levels by ferritin concentrations was much expected. Obviously, Hb concentration was not exclusively due to an iron deficit. The significant prediction is represented in bold; d.v., dependent variable; i.v., independent variable. The low R-square value in the presence of significance shows that even noisy, high-variability data can have a significant trend.

**Table 6 nutrients-11-01123-t006:** Prediction of haemodialysis adequacy by parameters of MICS and iron status.

Linear regression, Robust	Number of obs = 69		R-squared = 0.1726		
d.v. Kt/V	Coefficient	Standard Error	T	P > |t|	95% Conf. Interval
i.v. Transferrin	−0.0026856	0.0006328	−4.24	0.000	−0.0039459/−0.0014254
**Linear regression, Robust**	**Number of obs = 69**		**R-squared = 0.0019**		
d.v. Kt/V	Coefficient	Standard Error	T	P > |t|	95% Conf. Interval
i.v. Lymphocyte	0.0016172	0.0044794	0.36	0.719	−0.0073123/0.0105467
**Linear regression, Robust**	**Number of obs = 78**		**R-squared = 0.0119**		
d.v. Kt/V	Coefficient	Standard Error	T	P > |t|	95% Conf. Interval
i.v. Ferritin	0.0004338	0.0001691	2.57	0.013	0.0000963/0.0007713

Legend to Table 6. Transferrin is a known marker of inflammation, malnutrition, and iron status; ferritin, as marker of inflammation and iron status; lymphocyte count as a marker of inflammation and nutritional status. Bold represents the significant predictions; d.v., dependent variable; i.v., independent variable. The low R-square value in the presence of significance, as the case of the prediction of transferrin, shows that even noisy, high-variability data can have a significant trend.

**Table 7 nutrients-11-01123-t007:** Prediction of haemodialysis adequacy.

Equation	obs	Parms	RMSE	R-squared	F	P
Kt/V	68	4	0.2171343	0.2664	7.747195	0.0002
d.v. Kt/V	Coefficient	Standard Error	T	P > |t|	95% Conf. Interval	
i.v. CRP	−0.0017447	0.0008991	−1.94	0.057	−0.0035409/0.0000515	
i.v. Transferrin	−0.0026249	0.0007264	3.61	0.001	−0.0040762/−0.0011737	
i.v. Ferritin	0.0001767	0.0001692	1.04	0.300	−0.0001613/0.0005146	

Legend to Table 7. Multivariate regression. Parms, parameters; RMSE, root mean squared error (basically a measurement of accuracy). The more accurate model would have less error, as is evident in this output (i.e., 0.2). The significant “R-sq” should be noted. Bold represents which variable remained a significant predictor; d.v., dependent variable; i.v., independent variable.

**Table 8 nutrients-11-01123-t008:** Prediction of GNRI by transferrin, Kt/V, and CRP.

Equation	obs	Parms	RMSE	R-squared	F	P
GNRI	21	4	8.482248	0.4387	4.428626	0.0179
d.v. GNRI	Coefficient	Standard Error	T	P > |t|	95% Conf. Interval	
i.v. Transferrin	0.2204454	0.0604943	3.64	0.002	0.0928137/0.3480772	
i.v. Kt/V	13.20318	10.26218	1.29	0.215	−8.448127/34.85448	
i.v. CRP	−12.48504	10.78159	−1.16	0.263	−35.2322/−10.26212	

Legend to Table 8. The more accurate model would have less error (RMSE). Vice versa, it is not low in this output (i.e., 8.5), although significant as well as the R-squared. The significant predictor is highlighted in bold; d.v., dependent variable; i.v., independent variable.

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
