# Peer review of "Do Transferrin Levels Predict Haemodialysis Adequacy in Patients with End-Stage Renal Disease?"

_nutrients, 2019, doi:10.3390/nu11051123_

Round 1

Reviewer 1 Report

All comments were revised.

Author Response

Thank you very much indeed

Reviewer 2 Report

I have previously reviewed the manuscript and made comments and suggestions.

The authors have since modified the manuscript significantly.

Author Response

Thank you very much indeed

Reviewer 3 Report

In the revised manuscript and response letter, authors have successfully addressed most of the reviewer’s concerns. The reviewer has two comments regarding comment #1 and #5 as follows.

1. Authors should describe a rational explanation of that dialysis adequacy could be used as an indirect outcome in main text.

2. Even though it was a retrospective study, clinical study protocol has to receive ethical committee approval as long as it utilized patients’ data. 

Author Response

ANSWER TO REFEREE 3

Point 1

We have added the following sentence in the “Limitations” section: Finally, dialysis adequacy, evaluated as Kt/V, remains an open issue and whether it  could be used as an indirect outcome.

Point 2

We agree  that any manuscripts should include statements that provide a clear explanation as to why ethics approval and/or informed consent was not sought for a given study in a specific country or region. Statements such as “Ethics approval was not required for this study” or “Not applicable” do not provide a sufficient amount of detail; suitable reasons, and citations where applicable, should be provided.

For this reason we have previously clarified  our reasons in the Letter to Editor. Please, see below and furthermore, we point out that there are controversies about this topic.

We take the licence, with great respect, to add some passages drawn from literature.

Textually:

In conclusion, instutional ethics committee approval and informed consent are obligatory parts in all interventional studies (human or animal) without any doubt. However, recently published The National Code on Clinical Trials has declared that ethics approval is not necessary for real retrospective studies. After publishing this last version of ethics code, retrospective studies and cases which do not have ethics committee approval are acceptable to be reviewed in the most of medical journals though rarely it may not be the case.  

Kıraç FS. Is Ethics Approval Necessary for all Trials? A Clear But Not Certain Process. Mol Imaging Radionucl Ther. 2013;22(3):73–75. doi:10.4274/Mirt.80664

Textually:

Meanwhile, hospital-internal guidelines may impose stricter conditions than required by federal or cantonal law. Conversely, Swiss and European ethical texts may suggest greater flexibility and call for a looser interpretation of existing laws. 

Swiss Med Wkly. 2010 Jul 25;140:w13041. doi: 10.4414/smw.2010.13041. eCollection 2010.Retrospective research: What are the ethical and legal requirements? Junod V, Elger B.

We have provided the Editor with this explanation about the previous issues:

Dear Editor,

Authors did not seek informed consent or ethical committee approval

for their study, and provide the following justification for not

having done so:

1) The paper does not report on primary research. All data analysed

were collected as part of routine diagnosis and treatment.

2) Patients were diagnosed and treated according to national

guidelines and agreements. Testing blood as well as recording all

other variables included in our analysis is essential for confirming

diagnosis and classifying patients. It is done for each patient

without fail and as part of routine care, and is in no way an add-on

for purposes of research.

3) The paper does not report on the use of experimental or new protocols. The diagnostic protocol that we evaluated was the

canonical protocol.

4) The program was not set up as a study or research project.

5) Our analysis looked retrospectively at outcomes for a  cohort of patients treated by dialysis. This was done internally as part of an

evaluation, so as to improve our quality of care.

Best regards,

G.Tarantino M.D. and coll.

This manuscript is a resubmission of an earlier submission. The following is a list of the peer review reports and author responses from that submission.

Round 1

Reviewer 1 Report

This manuscript was about whether the transferrin levels predict dialysis adequacy in 2 patients with chronic renal failure? This study  evaluated which factor(s), if any, better predicted the Kt/V, weighing dialysis adequacy, among parameters.They found that levels of ferritin and transferrin, markers of both inflammation and iron status, did not impact on the eventual CRF-linked anemia in this population. The ferritin concentrations, both in males and females, were not particularly elevated, falling most  cases in normal range, indicating sufficient iron stores.  But there are some comments:

1.The manuscript must describe a technically sound piece of scientific research with data that supports the conclusions.

2.  In discussion section, implications to predict dialysis adequacy or nutrition status  should focus on the results findings.

3. Experimental design hadn't been considered in the execution of the research so that proper interpretation of the data can be made. The results and discussion need to be revised more clearly.

4. Do you assess serum iron: ferritin ratio? Hepcidin need to be measureed? About inflammation markers need to do more?

5. A lot of factors affect nutrition status and iron status need to be stated.

Author Response

REVISION for Nutrients-465183

Dear Editor,

We authors would like to thank the experts for their kind reviews, which allowed us to considerably improve this work. We truly enjoy this submission and mostly appreciated the reviewing  process and hope that these revisions will meet your approval and that the revised manuscript will be considered for acceptance in the Nutrients journal.

We look forward to hearing from you soon.   

G Tarantino, MD and Coll.

For your convenience we have highlighted the sentences  added or modified in the text as well as the related references.

Answer to Reviewer 1

This manuscript was about whether the transferrin levels predict dialysis adequacy in patients with chronic renal failure. This study  evaluated which factor(s), if any, better predicted the Kt/V, weighing dialysis adequacy, among parameters.They found that levels of ferritin and transferrin, markers of both inflammation and iron status, did not impact on the eventual CRF-linked anemia in this population. The ferritin concentrations, both in males and females, were not particularly elevated, falling most  cases in normal range, indicating sufficient iron stores.  But there are some comments:

Question (Q)1:The manuscript must describe a technically sound piece of scientific research with data that supports the conclusions.

Answer (A)1: We have made every effort to rigorously analysing data with proper statistical tools to drawn credible results and likely conclusions

Q 2:  In discussion section, implications to predict dialysis adequacy or nutrition status  should focus on the results findings.

A 2: We  have sincerely appreciated this suggestion and have stressed implications to predict  dialysis adequacy in the Discussion section, adding this sentence with related reference to the already existing one, i.e.,

….As possible strength of this study we emphasise that  detecting levels of transferrin to assess the dialysis quality combines in a unique test the evaluation of three mostly important factors of PEW……Furthermore, being both low dose of dialysis (Kt/V) and small body size found to be independent risk factors for mortality in chronic hemodialyzed patient, a parameter evaluating dialysis adequacy could help better manage these patients in order to improve their health.

Wolfe RAAshby VBDaugirdas JTAgodoa LYJones CAPort FK. Body size, dose of hemodialysis, and mortality. Am J Kidney Dis. 2000 Jan;35(1):80-8.

Q 3a: Experimental design hadn't been considered in the execution of the research so that proper interpretation of the data can be made.

A 3a: We are grateful to the reviewer for this suggestion. We have added in the conclusion that this observational study  with its preliminary results should be confirmed by larger prospective study.

Please, see below.

Q 3b:The results and discussion need to be revised more clearly.

A3b:  Thanking the reviewer, we have revised the Results and Discussion sections as suggested.

Q 4a: Do you assess serum iron: ferritin ratio? 

A4a:  Please, see below.

Q 4b: Hepcidin need to be measured? 

A 4b: We fully agree with this pertinent comment and have added a sentence in the limitation to study  about the evaluation of the iron status quoting the related articles, after the reference n 26, i.e., 

It should be stressed that we did not measured the serum hepcidin,  whose reduction of circulating concentrations by dialysis and/or rhEpo treatment may positively affect erythropoiesis.

Weiss GTheurl IEder SKoppelstaetter CKurz KSonnweber TKobold UMayer G.Serum hepcidin concentration in chronic haemodialysis patients: associations and effects of dialysis, iron and erythropoietin therapy.Eur J Clin Invest. 2009 Oct;39(10):883-90.

Nor was evaluated the iron: ferritin ratio that not only reflects the bioavailability of iron for the body, but also a healthy body composition.

Sabrina N, Bai CH, Chang CC, Chien YW, Chen JR, Chang JS. Serum iron: Ferritin ratio predicts healthy body composition and reduced risk of severe fatty liver in young adult women. Nutrients. 2017 Aug 4;9(8). 833. 

Q 4c: About inflammation markers need to do more?

A 4c: Also this observation is very adequate and we are grateful to the reviewer to let we deepen this aspect, expressing our opinion in the Discussion section.

Several markers were used for the detection of inflammatory reaction in patients with chronic renal disease.These markers can be used for the prediction of future cardiovascular events. Among the several parameters of inflammatory markers, e.g. TNF-alpha and Interleukin-6, serum CRP is well known and its advantages for the detection of inflammation and its predictor ability has been evaluated in several studies. In addition, their ability in predicting future atherosclerosis and effect of treatment are noteworthy. Furthermore, we dosed hs-CRP, a more precise marker, that seems to have a contribution in the development of cardiovascular complications.

 Li YZhong XCheng GZhao CZhang LHong YWan QHe RWang ZHs-CRP and all-cause, cardiovascular, and cancer mortality risk: A meta-analysis. Atherosclerosis. 2017 Apr;259:75-82.

Q 4d: A lot of factors affect nutrition status and iron status need to be stated.

A 4d: We thank the reviewer for this kind suggestion.

A paragraph labeled as Conclusions was added in which we clearly stated this aspect, i.e.,

Conclusions

The preliminary results  of this observational study, with some limits in its retrospective design,  show that  assessing the dialysis quality  by detecting levels of transferrin  combines in a unique test the evaluation of three mostly important factors of protein-energy wasting. Nevertheless, it should be  stressed that there is complexity of factors affecting both nutritional status and iron status. Larger prospective studies are needed to confirm our data.

Reviewer 2 Report

Tarantino et al., have performed a retrospective analysis on 78 Chronic renal failure patients, to evaluate the factors (nutrition, inflammation and Iron status) which better predicted the dialysis adequacy. The authors found that transferrin levels predicted the adequacy of Kt/v.

The manuscript needs extensive English language editing before can be reviewed.

There are so many grammatical mistakes, and it is very confusing right from the abstract to follow.

I suggest the authors resubmit after the editing.

Author Response

REVISION for Nutrients-465183

Dear Editor,

We authors would like to thank the experts for their kind reviews, which allowed us to considerably improve this work. We truly enjoy this submission and mostly appreciated the reviewing  process and hope that these revisions will meet your approval and that the revised manuscript will be considered for acceptance in the Nutrients journal.

We look forward to hearing from you soon.   

G Tarantino, MD and Coll.

For your convenience we have highlighted the sentences  added or modified in the text as well as the related references.

Answer to Reviewer 2

Tarantino et al., have performed a retrospective analysis on 78 Chronic renal failure patients, to evaluate the factors (nutrition, inflammation and Iron status) which better predicted the dialysis adequacy. The authors found that transferrin levels predicted the adequacy of Kt/v.

Question (Q)1: The manuscript needs extensive English language editing before can be reviewed.

There are so many grammatical mistakes, and it is very confusing right from the abstract to follow.

I suggest the authors resubmit after the editing.

Answer (A)1:

We would like to thank the Reviewer 2 for their appropriate suggestion and deeply regret for not having a strong command on the English.

However, we made sincere efforts to ameliorate the language with the help of a mother tongue expert.

Reviewer 3 Report

In this manuscript, authors demonstrated the associations between serum levels of transferrin and and adequacy of dialysis measured by Kt/V. Malnutrition and iron deficiency is important topics in dialysis fields. The subject of this study seems to be interesting. However, there are serious concerns in this study. The reviewer’s comments are described as following.

1. Dialysis adequacy determined by Kt/V is inappropriate as study outcome. Dialysis adequacy is therapeutic approach or intervention. The outcome should be some markers of malnutrition. Kt/V is a marker for performance of hemodialysis but not for patient prognosis. Although lower Kt/V generally results in higher mortality, higher Kt/V is not always associated with better prognosis and nutritional status. If authors would like to suggest that transferrin level was important to evaluate nutritional status in hemodialysis patients, the association must be adjusted by Kt/V.

2. Authors suggested that transferrin levels were superior to albumin or Geriatric Nutritional Risk Index (GNRI) to predict dialysis adequacy. However, this observational study design just shows the association between transferrin and Kt/V but cannot demonstrate the causality. Thus, whether transferrin level predicted better or worse Kt/V could not be demonstrated. Moreover, in this study, transferrin level was negatively correlated with Kt/V (the lower transferrin, the higher Kt/V), despite the well-known observation that low serum transferrin as well as hypoalbuminemia are more frequent in malnourished patients. Unless such complicated biological causality is clearly explained, the predictive ability of transferrin as authors suggested is not acceptable.

3. In this study, total number of study population was not shown. In other words, whether these 78 patients corresponded to the representative of the entire population remained unclear. Although inclusion and exclusion criteria are described in Methods, how many patients were excluded from the entire study population and why each patient was excluded must be shown. In addition, there were no information regarding how authors recruited the subjects (when, where, etc.) in this manuscript.

4. All laboratory data described in Methods (page 4, line 142-152) including transferrin should be listed in a new table as means ± standard deviations.

5. The name of ethical committee that approved this clinical study and the approval number assigned by the committee must be described in Methods.

Round 2

Reviewer 1 Report

All comments are revised.

Reviewer 2 Report

I have reviewed the manuscript previously, and unfortunately, the submission still needs an English language review. Please furnish a certificate to prove the same with the resubmission.

Here are my comments and suggestions,

A.Abstract:

-Please mention in the abstract whether the studied patients are Hemo or peritoneal dialysis patients or both.

-Please use the term End-stage renal disease instead of Chronic renal failure, where ever necessary in the manuscript, which appropriately reflects the study population.

B. Introduction:

- Citation 1 is for the patients on the Peritoneal dialysis. Authors need to change it.

-Page 2 of 12. 2nd paragraph. The authors wrote ‘Recently, experts have labeled the uremic malnutrition as protein-energy wasting (PEW).’ This and subsequent details need a citation.

Under (c), the authors wrote, measures of muscle mass such as total muscle mass, mid-arm muscle circumference, and creatinine appearance.’ What is Creatinine appearance?

-Page 3 of 12, paragraph 5. The authors wrote ‘Still, lasting controversies about the correct use of parameters assessing nutritional status, inflammation response, iron status as well as weighing dialysis adequacy, we aimed at evaluating which factor(s), if any, better predicted the Kt/V.’

The sentences like this in the manuscript, need English editing as it's confusing.

C. Methods:

-Population. ‘We analyzed data from 78 CRF patients not.’ Please use the term ESRD instead of CRF.

The authors wrote, ‘symptomatic angor.’ What does that mean?

-The authors wrote ‘without a history or instrumental assessment of ischemic heart disease in the same period.  Changes in the ECG responses to exercise (treadmill heart test) were assessed to all subjects to exclude cardiac ischemic signs.’

Do they mean interventional assessment of the ischemic heart disease?

-The authors wrote that the data were collected retrospectively from patient records. If it so, why and how the patients were tested for the clinical markers or laboratory values which were used to exclude patients as described in the exclusion criteria.

-How did the authors recruit the patients? Who and how they collected data? Consent? IRB? Basic demographics of the patients? The type of dialysis? How long were the patients on the dialysis?

- Why have they not included the Cholesterol which is a marker in the diagnostic criteria used in the Protein-energy wasting (PEW)?

Again, the discussion is poorly written, and authors should include an English language expert and an English speaking nephrologist to edit the manuscript before the resubmission.